# Burden and determinants of scabies in Ethiopian school age children: A systematic review and meta-analysis with public health implications

Lidetu Demoze[1]*, Fetlework Gubena[2], Eyob Akalewold[2], Helen Brhan[2], Kassaw Chekole Adane[3], Tigist Kifle[2], Natnael Gizachew[4], Zemichael Gizaw[1], Amensisa Hailu Tesfaye[1,5], Gelila Yitageasu[1]

1 Department of Environmental and Occupational Health and Safety, Institute of Public Health, College of Medicine and Health Sciences, University of Gondar, Gondar, Ethiopia, 2 Department of Epidemiology and Biostatistics, Institute of Public Health, College of Medicine and Health Sciences, University of Gondar, Gondar, Ethiopia, 3 Department of Environmental and Occupational Health and Safety, College of Medicine and Health Sciences, Wollo University, Dessie, Ethiopia, 4 School of Public Health, College of Health Science and Medicine, Dilla University, Dilla, Ethiopia, 5 Institute for Sustainable Futures, University of Technology Sydney, Ultimo, New South Wales, Australia

* lidetudemoze12@gmail.com

**Data Availability Statement:** All data generated or analyzed during this study are included in this

## Abstract

### Background

Scabies is a major global health problem, affecting an estimated 200 million people worldwide and causing more than 455 million new cases annually. It is one of the most prevalent dermatological conditions in developing countries and vulnerable populations, such as school-age children. In Ethiopia, findings regarding the prevalence and associated factors of scabies among school age children have been irregular and inconsistent. In addition, no previous systematic reviews or meta-analyses have been conducted in Ethiopia. Therefore, this systematic review and meta-analysis was conducted to estimate the pooled prevalence of scabies and their associated factors among school age children in Ethiopia.

### Methods

We conducted a systematic literature search (April 9–12, 2024) to identify studies on scabies incidence and associated factors in Ethiopian children among school age children. Published and peer reviewed articles were searched in electronic databases such as Medline/PubMed, Embase, Scopus, Science Direct, and Epistemonikos. In addition, other search methods, such as Google Scholar, Ethiopian University and Research Institutional Repository, and Google manual searches were also conducted. All papers published until 12 April 2024 were considered. We conducted a systematic review and meta-analysis using the Preferred Reporting Items for Systematic Reviews and Meta-Analyses (PRISMA) guidelines. Only studies that met the predefined inclusion criteria were included. Quantitative methods were employed to analyze the data, and heterogeneity across studies was assessed. Additionally, publication bias was evaluated via a funnel plot and Egger's regression test.

published article and its supplementary information files.

**Funding:** The author(s) received no specific funding for this work.

**Competing interests:** The authors have declared that no competing interests exist.

Publication bias was assessed via funnel plots and Egger's regression test. The protocol for this review has been registered with PROSPERO (ID: CRD42024532037).

## Results

A comprehensive systematic review of 1,144 studies identified 16 studies that met the inclusion criteria were included in this systematic review and meta-analysis. The pooled prevalence of scabies among school age children in Ethiopia, based on the 16 included studies, was 21.1% (95% CI: 15.0%, 27.2%). A family history of scabies, knowledge about scabies, sharing a bed, sleeping with a scabies-ill person, sharing a cloth, contact with a person who has symptoms of scabies, and sleeping place are some of the factors significantly associated with scabies among school age children in Ethiopia.

## Conclusions

The pooled prevalence of scabies among school age children in Ethiopia was high. Sociodemographic, water, sanitation, and hygiene factors were associated with scabies. A multipronged approach is recommended to address scabies in Ethiopian school age children. Strengthening collaboration among the education, water, and health sectors would promote a coordinated response. Such interventions have the potential to reduce the prevalence of scabies in this vulnerable population significantly.

## Introduction

Scabies stands as one of the most prevalent dermatological conditions, constituting a significant portion of skin diseases in developing nations [1]. Scabies is caused by the ectoparasite *Sarcoptic scabiei* var. *hominis*, and transmission generally requires skin-to-skin contact [2]. In 2017, the World Health Organization acknowledged scabies as one of the neglected tropical diseases [3]. It is estimated that neglected tropical diseases (NTDs) affect more than 1 billion people, with scabies alone impacting more than 200 million individuals at any given time and over 400 million people each year [4, 5]. Scabies was responsible for 0.21% of Disability-Adjusted Life Years from all conditions studied by GBD 2015 worldwide [6]. Scabies prevalence varied widely across regions, with rates ranging from 0.2% to 71.4%, exceeding 10% in all regions except Europe and the Middle East, and reaching the highest levels in the Pacific and Latin American regions, particularly among children [7].

The prevalence of scabies in Africa is estimated to be 4.7% in Nigeria and 70% in Ghana [8]. In addition, studies indicate that the prevalence of scabies among school age children varies across studies in Africa (2.0%, 2.9% and 65%, respectively) [9–11]. A previous systematic review and meta-analysis estimated that the overall prevalence of scabies was 14.5% (95% CI: 1.5, 27.6%) in the general population in Ethiopia [12]. However, epidemiological studies indicate that the prevalence of scabies among school age children in Ethiopia varies across studies. The estimated prevalence of scabies among school age children in Ethiopia varies by region, ranging from 5.3% in the Southern Nations, Nationalities, and People's Region, 10.82% in the Amhara Region, 16.6% in the Sidama Region, and 12.93% in the Tigray Region, to as high as 53% in Oromia [13–17].

There was also methodological variation across the studies to ascertain scabies among school age children. Scabies and other infectious skin diseases are common among school age

children because of close contact between classmates. Several factors, such as age, sex, parental education, occupation, cloth and bed-sharing, family size and personal hygiene, are associated with the development of scabies among school-age children [18–21].

To date, no comprehensive systematic review or meta-analysis has synthesized the available evidence on the prevalence and determinants of scabies among school age children in Ethiopia. This study provides a novel perspective on the burden of scabies among school age children in Ethiopia. Furthermore, this systematic review and meta-analysis incorporates multiple recent Ethiopian studies focusing on this population. This study aims to fill this gap by conducting a systematic review and meta-analysis to estimate the pooled prevalence of scabies and identify the key sociodemographic, environmental, and behavioural factors associated with the disease in this population. The findings from this study will provide valuable insights to guide the design and implementation of scabies control programs, ultimately improving the health and well-being of school age children in Ethiopia.

In addition, this approach can highlight variations in prevalence across categories and offer a detailed understanding of the epidemiology of scabies among school age children in Ethiopia [22]. Furthermore, it will provide evidence-based interventions through comprehensive data that will enable us to make well-informed decisions. The goal of this research was to conduct a systematic review of Ethiopian studies examining the prevalence of scabies and associated factors among school age children.

## Methods and materials

### Protocol and registration

The protocol for this review was registered in the International Prospective Register of Systematic Reviews (PROSPERO) (record ID: CRD42024532037). Available at https://www.crd.york.ac.uk/prospero/#myprospero.

### Search strategies

The systematic review was developed according to the Preferred Reporting Items for Systematic Reviews and Meta-Analyses (PRISMA) guidelines [23], and the review procedure was reported via the PRISMA-P 2020 checklist [24] (S1 File). Articles were searched in electronic databases such as Medline/PubMed, Embase, Scopus, Science Direct, and Epistemonikos. The rationale for selecting these specific databases was to ensure comprehensive coverage of published, peer-reviewed articles. In addition, other search methods, such as Google Scholar, Ethiopian University and the Research institutional repository, and Google manual searches were conducted from April 09 to April 12/2024. The Google Scholar search was manually conducted using specific keywords related to scabies prevalence among school age children in Ethiopia. All papers published until 12 April 2024 were considered (S2 File). MeSH terms and entry terms were used with the following keywords: "prevalence", "scabies", "school age children", "determinant factor", "associated factor" and "Ethiopia". The search terms were used separately and in combination with Boolean operators such as "OR" or "AND".

### Eligibility criteria

**Inclusion criteria.** *Study area*. Studies conducted in all Ethiopian regional states.

*Population*. School age children.

*Publication conditions*. Peer-reviewed journal articles published before April 12/2024.

*Publication year*. No restriction.

*Study design.* All observational studies (i.e., cross-sectional, case-control and cohort) reporting the prevalence of scabies in school age children were eligible for this review.

*Language.* Only articles reported in the English language were considered. English was the predominant language of international scholarly communication. Many Ethiopian researchers publish their work in English to reach a global audience.

## Exclusion criteria

Studies that did not report the prevalence of scabies among school age children; did not have full-text articles following three or more email contacts with the corresponding authors; or were systematic reviews, qualitative studies, letters, conference abstracts, short communications, commentaries and case reports were excluded. Conference abstracts and non-peer-reviewed sources were excluded to ensure the inclusion of high-quality, rigorously vetted information that met established scientific standards. Conference abstracts often lack detailed methodology and comprehensive results, which are essential for accurate data interpretation, whereas non-peer-reviewed sources may not have undergone critical evaluation by experts, potentially introducing bias or inaccuracies.

## Outcome of interest

The primary objective of this study was to determine the pooled prevalence of scabies among school age children in Ethiopia. The prevalence was calculated by dividing the number of school-age children with scabies by the total number of school age children in the study and then multiplying the result by 100. For the study's secondary objective (factors associated with scabies), we collected data on factors considered to be related to scabies in the literature. Data from the primary studies were collected in the form of two-by-two tables for the analysis of scabies-related factors, and the odds ratio (OR) was calculated to determine the relationship between each of the explanatory variables and scabies.

## Study selection process

Endnote reference manager software version 20.5 [25] was used to collect and organize the search results and to remove duplicate articles. Duplicate studies were removed. Two investigators (LD and GY) independently evaluated the articles for eligibility, examining titles, abstracts, and full texts against established inclusion and exclusion criteria. The screened articles were then compiled by three investigators (FG, EA, ZG, and HB). When the authors disagreed during the selection process, they resolved the issue through evidence-based discussion and the involvement of other investigators (KC, ZG, TK, NG, and AHT).

## Data extraction and management

The data were extracted from each study in a standardized format. Three reviewers (LD, GY, and KC) independently extracted the data via a spreadsheet, and any disagreements were resolved through discussion and additional review by other team members (TK, EA, NG, FG, ZG, AHT and HB). The extracted information included the name of the first author, publication year, study period, country, design, sample size, number of scabies cases, prevalence of scabies, associated factors, and measures of association (ORs). To maintain the robustness (handling missing data) of the present systematic review and meta-analysis, we conducted sensitivity analyses, performed subgroup analyses, assessed publication bias, followed standard data extraction methods and performed transparent reporting.

### Risk of bias assessment

A full-text review of studies was performed before the inclusion of studies in the final meta-analysis via the "Newcastle-Ottawa Scale (NOS)" quality appraisal tool (S3 File) adapted for both cross-sectional and case-control studies [26], with a total score of 10 for cross-sectional studies and 9 for case-control studies. Cross-sectional studies with scores of 9–10 were considered very good, studies with scores of 7–8 were considered good, studies with scores of 5–6 points were considered satisfactory, studies with scores of 0–4 were considered unsatisfactory, and studies with no standard score classification for case-control studies were given scores ranging from 0–9. The components of quality assessment for cross-sectional studies include representativeness of the sample, sample size and nonrespondents, ascertainment of the exposure (risk factor), comparability of subjects, confounding factors, statistical tests and assessment of outcomes. The components of quality assessment for a case-control study include case definition, representativeness of the cases, selection of controls, the definition of controls, comparability of cases and controls, ascertainment of exposure, ascertainment for cases and controls and nonresponse rate. The independent quality assessment of the studies was reviewed by LD and GY. Disagreements among the reviewers during the quality assessment were resolved via discussion.

### Data processing and analysis

The extracted data were exported from an Excel spreadsheet to STATA version 17 for further analysis (S4 File). To ascertain study heterogeneity, we computed the $I^2$ statistic, which expresses the proportion of overall variation across studies attributable to heterogeneity as opposed to chance. Low, moderate, and high heterogeneity were represented by values of 25%, 50%, and 75%, respectively [27]. Significant heterogeneity between the studies was revealed by the test statistic ($I^2$ = 97.65%, P < 0.00). As a result, a random effects model was used to evaluate the pooled prevalence of scabies among school age children because it accounts for the variability/heterogeneity between studies. Prevalence and odds ratios with 95% confidence intervals were calculated via DerSimonian-Laird weights [28]. To minimize the heterogeneity between the point estimates of the studies, subgroup analysis was conducted based on the study setting and study design. Furthermore, sensitivity analysis was performed to identify the impact of individual studies (extreme outliers) on the pooled estimate. Sensitivity analysis was conducted via a leave-one-out approach, where each study was excluded one at a time to observe the impact on the overall effect size. This method allowed for the assessment of how the removal of individual studies influenced the robustness and reliability of the results. Publication bias was detected via both visual and statistical methods (funnel plot and Egger's test, respectively) at a p-value below 0.05 [29]. To assess the associations between scabies and potential risk factors, odds ratios with 95% confidence intervals were calculated. A p-value of 0.05 was used as the threshold for statistical significance.

### Results

A total of 1144 articles were retrieved via electronic database searches: Medline/PubMed, Embase, Scopus, Science Direct, and Epistemonikos. Other search methods include Google Scholar, Ethiopian University and Research Institutional Repository, and Google manual search. Seven hundred thirty-one articles were excluded because of duplication, and 383 articles were removed because they were not related to the title and abstract of our study. 14 more articles were excluded because their study population was not school age children. Finally, 16 articles were included in the systematic review and meta-analysis (Fig 1).

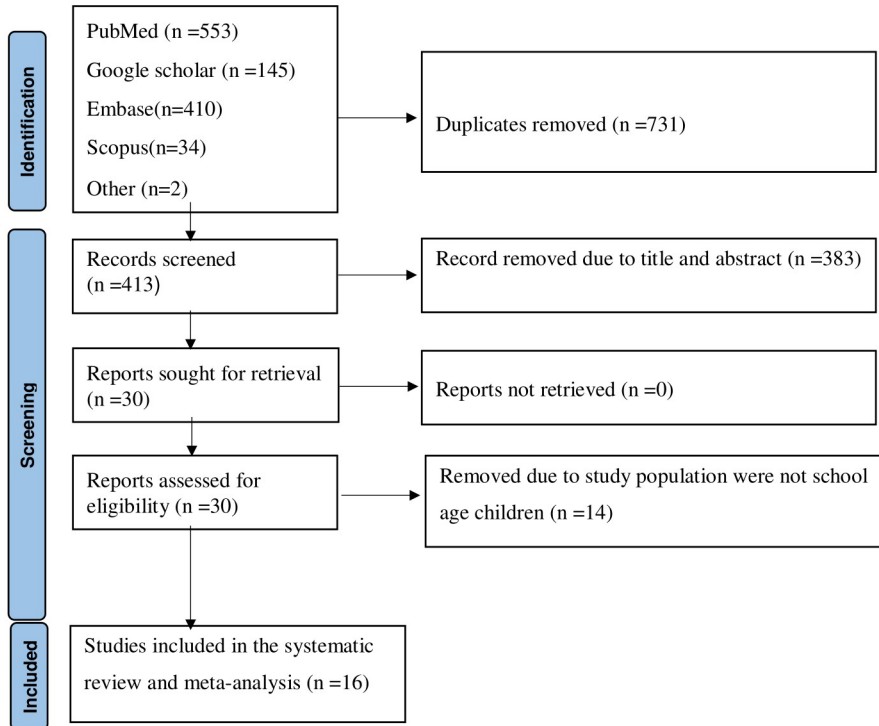

**Fig 1. PRISMA 2020 flow diagram of the included studies to estimate the prevalence of scabies and associated factors among school age children in Ethiopia.**

## Characteristics of the studies

A total of 12 cross-sectional and 4 case-control studies with 8822 sample sizes were included in this systematic review and meta-analysis. Six studies were from the Amhara Region [15, 30–34], four studies were from the Oromia Region [14, 35–37], three studies were from the SNNP Region [13, 38, 39], two studies were from the Sidama Region [16, 40], and one study was from the Tigray Region [17]. Seven studies were community-based, and the remaining (9 studies) were institution-based. The sample size ranged from 96 in the Gondar city Amhara Region [34] to 861 in the Wonago district SNNP Region [13]. With the exception of one study [34], the entire set of included studies included both sexes [13–17, 30–33, 35–40]. Before any analysis, the articles were all reevaluated by impartial assessors, and the studies passed the quality fit test (Table 1).

## Meta-analysis

**Pooled prevalence of scabies among school age children in Ethiopia.** The pooled prevalence of scabies among school age children in Ethiopia was 19% (95% CI: 14, 23). The lowest prevalence was reported in SNNP, at 5.3% (95% CI: 4, 7) [13], and the highest was reported in Oromia, at 53% (95% CI: 48, 59) [14]. The $I^2$ test revealed heterogeneity among the included studies ($I^2$ = 97.65%, p-value < 0.00) (Fig 2). Subgroup and sensitivity analyses were therefore carried out to pinpoint the potential sources of heterogeneity.

**Subgroup analysis.** The subgroup prevalence of scabies was estimated by considering the region, study design and study setting (community and institution). In the subgroup analysis, the pooled prevalence of scabies was highest in the Oromia region at (95% CI: 11, 37), and

**Table 1. The characteristics of the articles included in the systematic review and meta-analysis of the prevalence and associated risk factors for scabies among school age children in Ethiopia, 2024.**

| Author (s) and Publication Year | Region | Study design | Sample size | Response rate | Prevalence | Quality score |
|---|---|---|---|---|---|---|
| Tefera Haile et al.,2020 [30] | Amhara | CS | 583 | 96.4 | 23.8 | Good |
| Abayneh Tunje et al.,2018 [38] | SNNP | CS | 825 | 97.6 | 16.4 | Good |
| Yahya Kemer et al.,2022 [35] | Oromia | CS | 447 | 97 | 11 | Good |
| Desta Marmara et al.,2022 [16] | Sidama | CS | 590 | 97.7 | 16.6 | Good |
| Bisrat Misganaw et al., 2022 [15] | Amhara | CS | 850 | 98.04 | 10.82 | Good |
| Sindayo Tefera et al., 2020 [17] | Tigray | CS | 495 | 100 | 12.93 | Good |
| Stephen L. Walker et al.,2017 [40] | Sidama | CS | 343 | 100 | 5.5 | Unsatisfactory |
| Gemechu Ararsa et al.,2023 [36] | Oromia | CS | 457 | 99.13 | 19.26 | Good |
| Henok Dagne et al.,2019 [31] | Amhara | CS | 494 | 91.84 | 9.3 | Good |
| Tarkie Abebe et al.,2023 [32] | Amhara | CS | 622 | 98 | 8.8 | Good |
| Hiwot Hailu Amare et al. [13] | SNNP | CS | 861 | 100 | 5.3 | Good |
| Yohannes Luluet al.,2017 [37] | Oromia | CS | 828 | 100 | 13.8 | Very good |
| Melat Wodaje et al.,2020 [33] | Amhara | CC | 300 | 100 | 33.3 | 9 |
| Kefele Ejigu et al.,2019 [39] | SNNP | CC | 711 | 98 | 33.3 | 8 |
| Yassin Zeyneba et al.,2017 [34] | Amhara | CC | 96 | 100 | 33.3 | 8 |
| Eden Gebre [14] | Oromia | CC | 320 | 94.1 | 53.1 | 7 |

CC = Case–control, CS = Cross-sectional, SNNP = Southern, Nations, Nationalities and People's

lowest in the Sidama region at 10% (95% CI: 8, 12) (Fig 3). The overall pooled prevalence of scabies based on the study design was 13.0% (95% CI: 10, 16) for the cross-sectional study and 38.0% (95% CI: 29, 48) for the case–control study (Fig 4). Additionally, the overall pooled prevalence of scabies based on the study setting was 20.0% (95% CI: 13, 28) for the

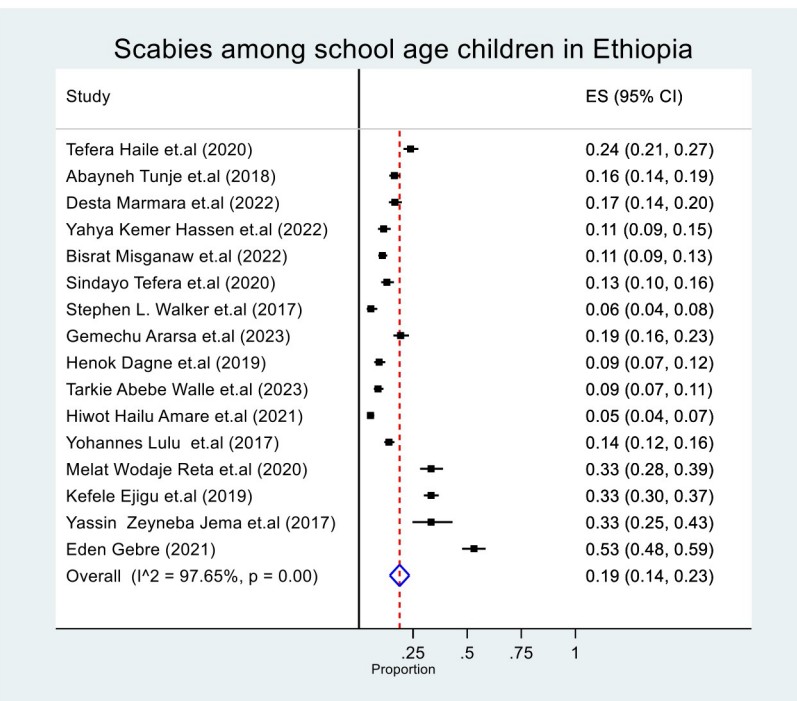

**Fig 2. Forest plot of the pooled prevalence of scabies among school age children in Ethiopia.**

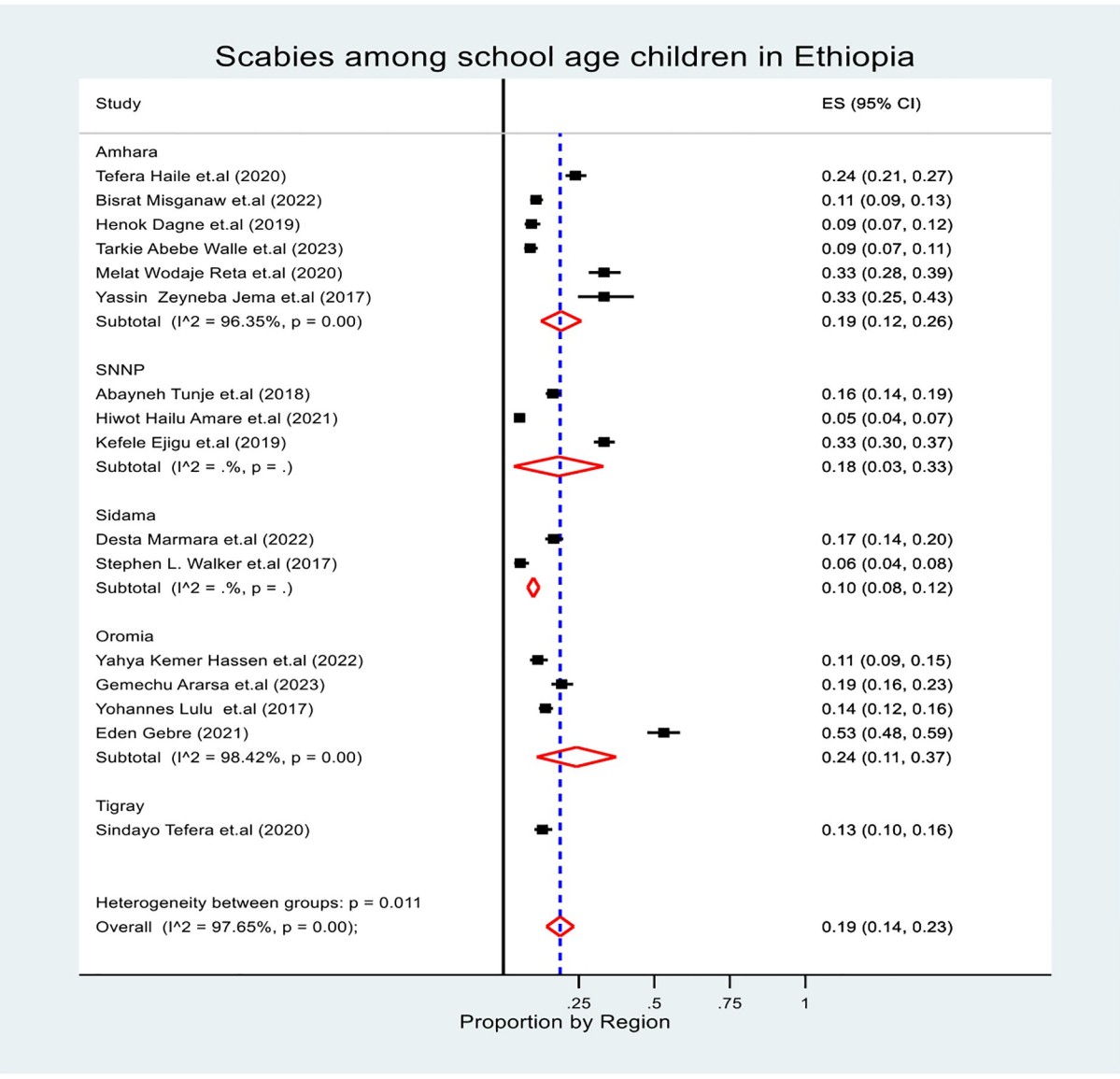

**Fig 3. Forest plot of the pooled prevalence of scabies among school age children in Ethiopia by region.**

community-based study and 17% (95% CI: 12, 23) for the institution-based study (Fig 5). In general, during subgroup analysis, the I² test for both subgroup analyses revealed heterogeneity across the studies.

**Sensitivity analysis.** The impact of each study on the pooled prevalence was assessed through a sensitivity analysis. However, the analysis revealed that no single study had a substantial effect on the overall estimate in the meta-analytic model (Fig 6).

**Small study effect test (assessment of publication bias).** All sixteen included studies were assessed for publication bias (small study effect). The presence of a possible small study effect was checked by using a funnel plot and Egger's test. Accordingly, the funnel plot (Fig 7) showed an asymmetric distribution and presented evidence of a small study effect. In addition, the results of Egger's test indicated that there was evidence of publication bias (small study effect) (P value = 0.000) (Fig 8).

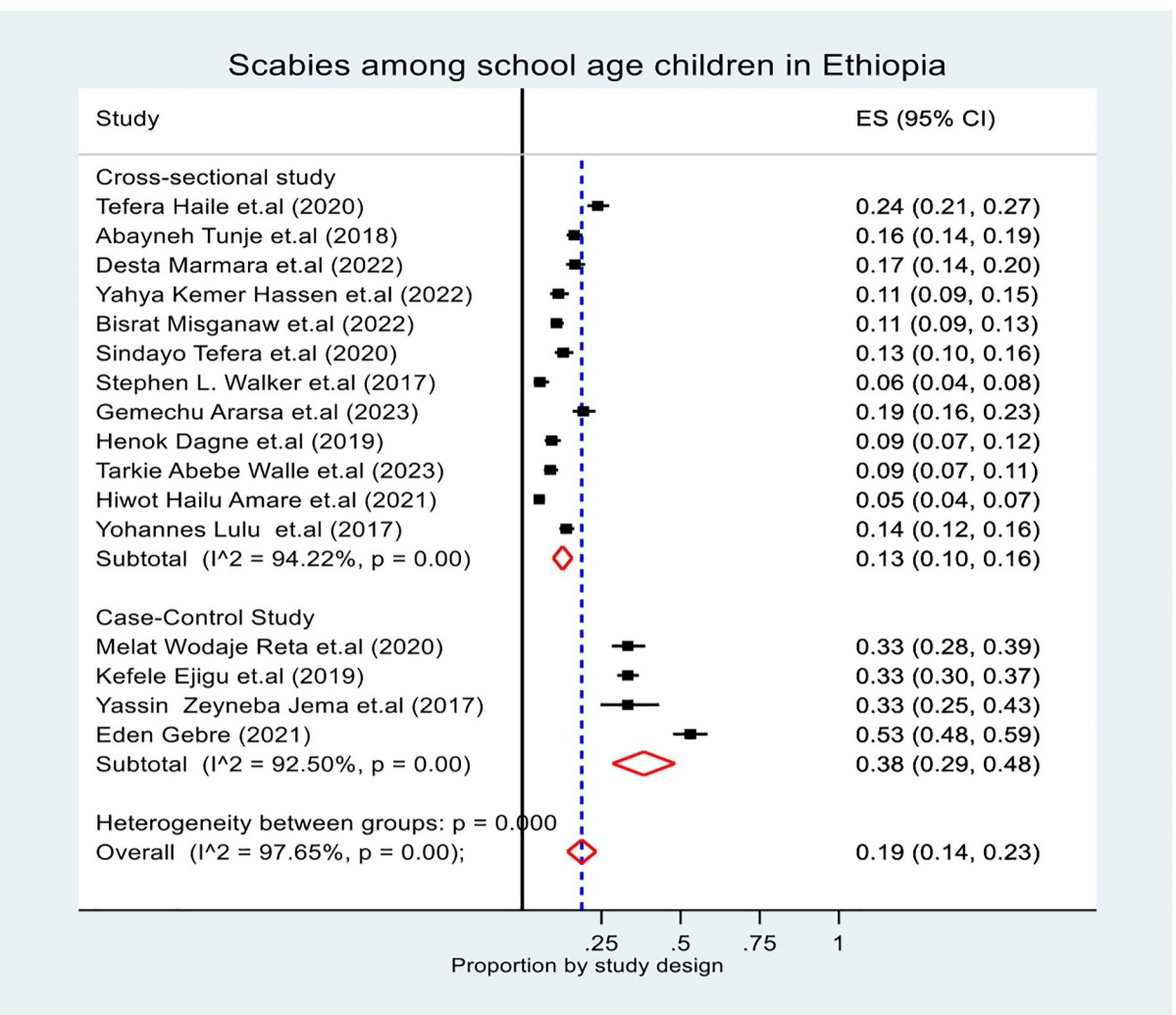

**Fig 4. Forest plot of the pooled prevalence of scabies among school age children in Ethiopia by study design.**

**Trim and fill analysis.** Publication bias was observed during Egger's test and a funnel plot [41]. By considering publication bias, trim-and-fill analysis was performed to account for publication bias [42]. The trim-and-fill analysis results (Figs 9 and 10) revealed that the prevalence of scabies among school age children was 21.12% (95% CI: 15.0, 27.2).

**Factors associated with scabies among school age children in Ethiopia.** Factors associated with scabies were identified based on the pooled effect of two or more studies. Sex, education of father, fingernails cut, family history with scabies, knowledge of scabies, sharing a bed, sleep with scabies ill person, frequency of washing cloth, sharing of cloth, contact with a person who had scabies symptoms, sleeping place, frequency of bath, using a detergent for washing, travel to epidemic areas and water source were identified as factors associated with scabies among school age children in Ethiopia.

The pooled effect of our review analysis showed the odds of having scabies were 1.44 times higher among male school age children than females [POR: 1.44; 95% CI (1.03, 2.00)]. Fathers of school age children who were illiterate their children's were 2.10 times more likely to experience scabies compared to those who had literate fathers [POR: 2.10; 95% CI (1.31, 3.37)].

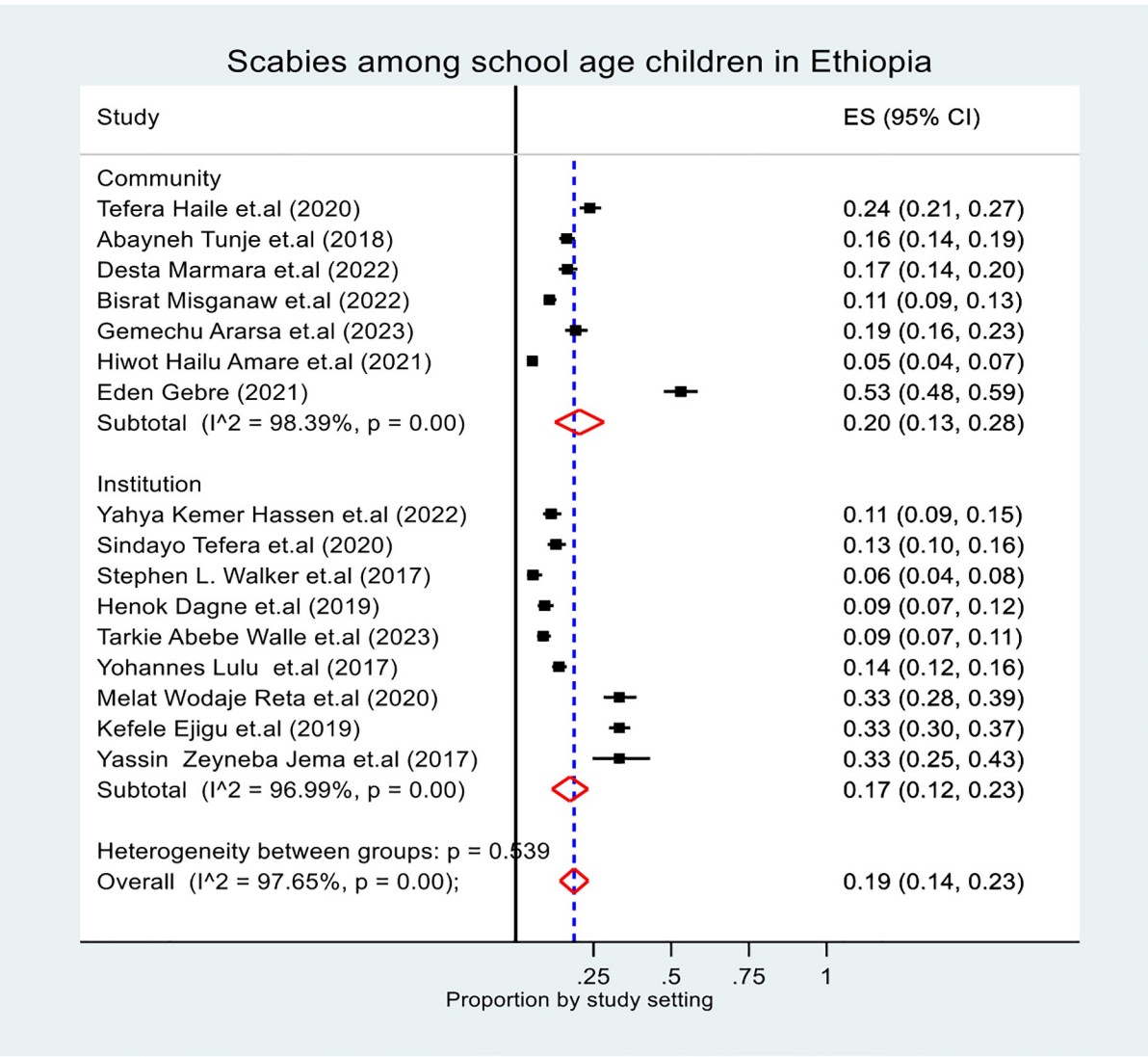

**Fig 5. Forest plot of the pooled prevalence of scabies among school age children in Ethiopia by study setting.**

School age children who didn't cut their fingernails were more likely to have scabies compared to those who did cut their fingernails [POR: 3.26; 95% CI (1.31, 8.10)]. As well, the pooled effect of our analysis indicated that the likelihood of suffering from scabies was increased sevenfold among school age children who had a family history of scabies as compared to their counterparts [POR: 7.25; 95% CI (4.85, 10.85)]. School age children who had poor knowledge about scabies were 3.80 times more likely to develop scabies compared to school age children who had good knowledge about scabies [POR: 3.80; 95% CI (2.29, 6.31)]. The odds of having scabies were 3.49 times higher among school age children who shared a bed than those who did not [POR: 3.49; 95% CI (2.12, 5.73)]. School age children who slept with a scabies ill person were 3.49 more likely to have scabies compared to those who did not [POR: 3.49; 95% CI (1.78, 6.83)]. This study showed that school age children who did not wash their clothes frequently were 2.39 times more likely to experience scabies compared to those who did [POR: 2.39; 95% CI (1.51, 3.77)]. The odds of having scabies were 3.18 times higher among school

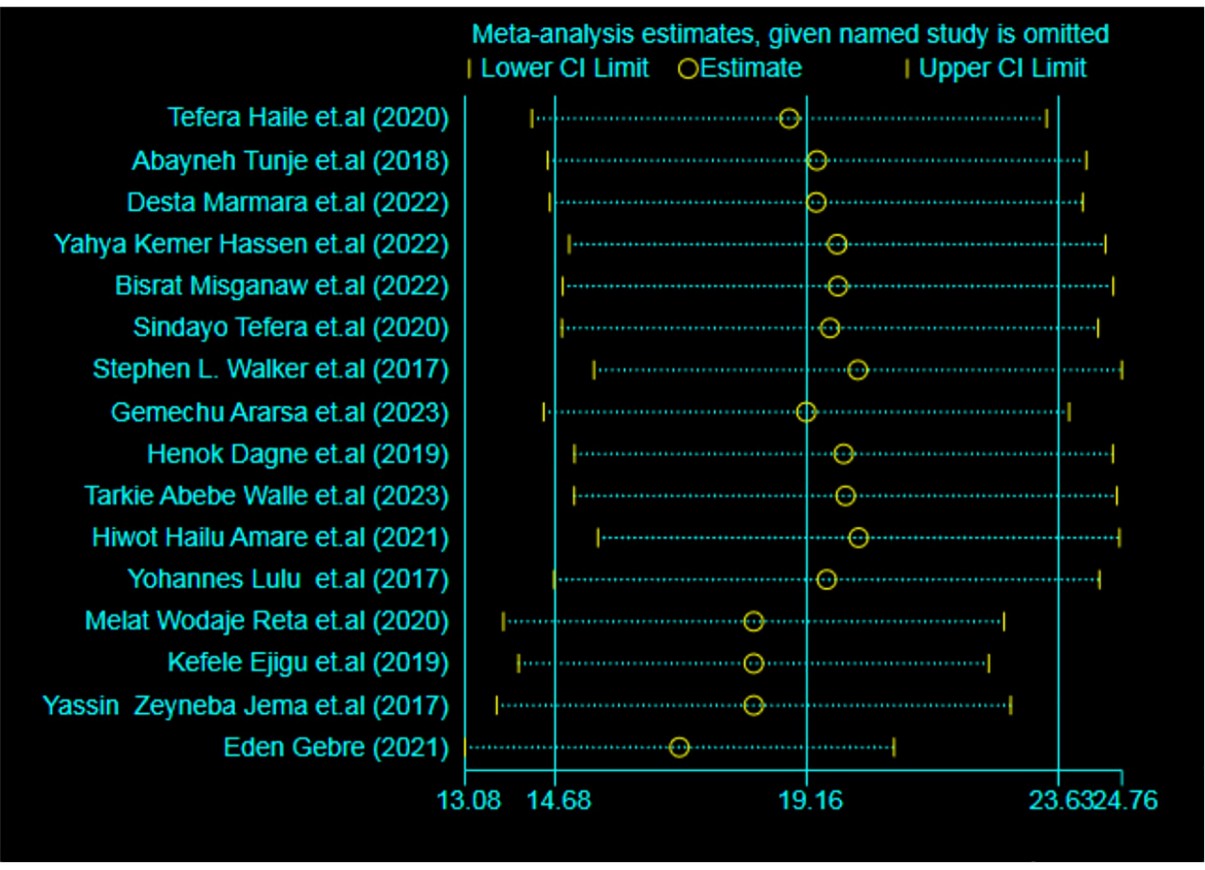

**Fig 6. Sensitivity analysis to examine the effect of a single study on the pooled result.**

age children who shared cloth than those who did not [POR: 3.18; 95% CI (1.17, 8.64)]. School age children who had contact with persons who had scabies symptoms were 5.78 times more likely to have scabies compared to those who did not [POR: 5.78; 95% CI (3.73, 8.97)].

The odds of having scabies were 5.82 times higher among school age children who slept on the floor than those who did not [POR: 5.82; 95% CI (2.80, 12.07)]. Higher odds of having scabies were observed among school age children who did not take baths frequently than those who took baths frequently [POR: 1.88; 95% CI (1.16, 3.03)]. In addition, school age children households' who didn't use detergents for washing, their school age children were 2.28 more likely to have scabies compared to those who used detergents [POR: 2.28; 95% CI (1.54, 3.37)]. Furthermore, school age children who travelled to the scabies epidemic area were 2.93 times more likely to have scabies compared to those who did not [POR: 2.93; 95% CI (1.33, 6.43)]. Finally, higher odds of having scabies were observed among school age children who use unimproved water sources than those who use improved water sources [POR: 3.04; 95% CI (1.35, 6.86)]. However, there was no significant pooled effect size observed between age, education of mothers, family size, previous history of scabies infection, frequency of changing clothes, occupation with scabies among school age children in Ethiopia in our analysis (Table 2).

The meta-analysis on scabies prevalence among school age children in Ethiopia showed a pooled prevalence of 21.1%, with significant regional variability, and identified associated risk factors, including a family history of scabies, knowledge about scabies, sharing a bed, sleeping with scabies ill person, sharing a cloth, contact with a person who has symptoms, of scabies, and living in a sleeping place.

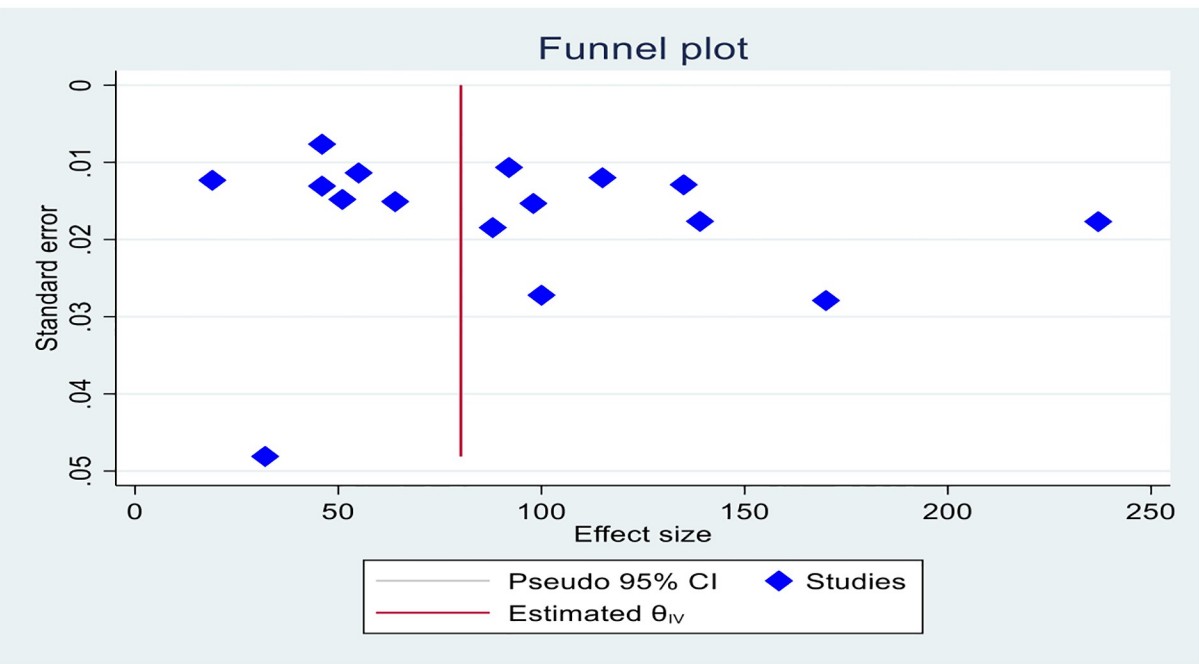

**Fig 7. Funnel plot of the sixteen studies included in the meta-analysis of scabies among school-age children in Ethiopia, 2024.**

## Discussion

This systematic review and meta-analysis aimed to synthesize the available evidence on the pooled prevalence of scabies and its determinants among school age children in Ethiopia. Despite the numerous primary studies conducted on the burden of scabies in this population,

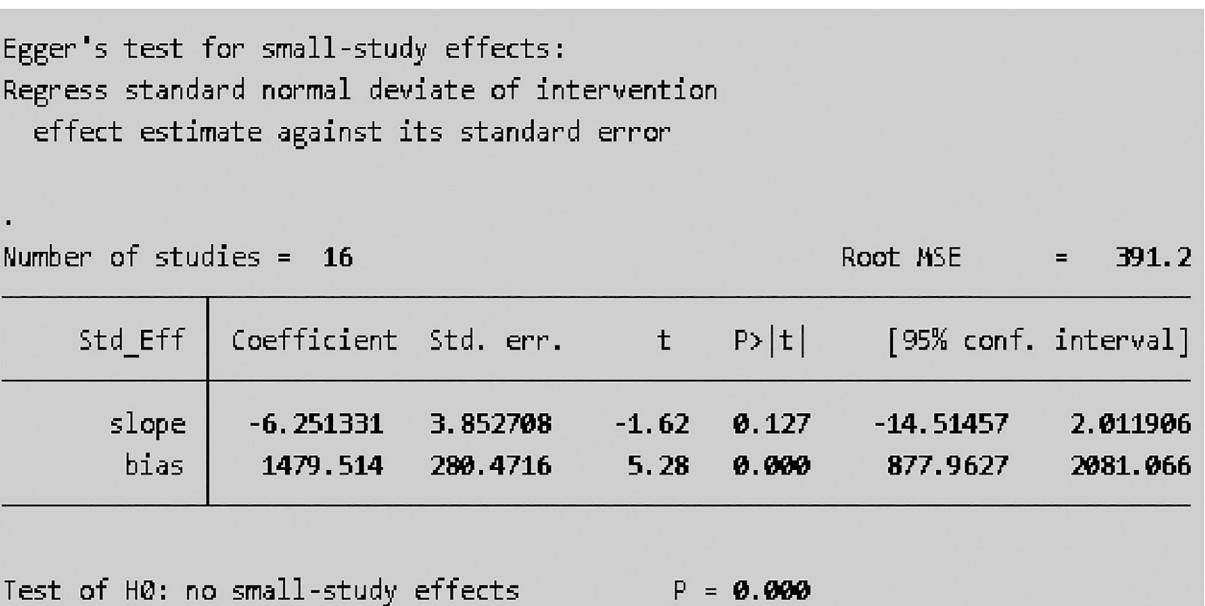

**Fig 8. Egger's test of the sixteen studies included in the meta-analysis of scabies among school-age children in Ethiopia, 2024.**

```
Nonparametric trim-and-fill analysis of publication bias
Linear estimator, imputing on the right

Iteration                              Number of studies =      18
  Model: Random-effects                          observed =      16
 Method: DerSimonian-Laird                         imputed =       2

Pooling
  Model: Random-effects
 Method: DerSimonian-Laird
```

| Studies | Effect size | [95% conf. interval] | |
|---|---|---|---|
| Observed | 19.157 | 14.681 | 23.633 |
| Observed + Imputed | 21.122 | 15.042 | 27.203 |

**Fig 9. Trim-and-fill analysis of the sixteen studies included in the meta-analysis of scabies among school-age children in Ethiopia, 2024.**

the findings have been inconsistent, likely due to heterogeneity in study designs, settings, and sample sizes. To our knowledge, this is the first comprehensive systematic review and meta-analysis to estimate the overall prevalence of scabies and identify the key sociodemographic, environmental, and behavioural factors associated with this disease among school age children in Ethiopia. The results of this study provide valuable insights that can inform policymakers and public health practitioners in Ethiopia. Estimating the pooled prevalence of scabies in this vulnerable population is crucial for guiding the allocation of resources, designing targeted intervention strategies, and monitoring progress toward scabies control and elimination efforts. Furthermore, the identification of risk factors associated with scabies can inform the development of evidence-based prevention and management programs, ultimately improving the health and well-being of school-age children in Ethiopia. The findings from this review may also have broader relevance for other resource-limited settings associated with the burden of scabies among school age populations.

This meta-analysis revealed that the prevalence of scabies among school age children in Ethiopia was 21.1% (95% CI: 15.0, 27.2). The current pooled prevalence of scabies among school age children is consistent with findings from two cross-sectional studies conducted in India and one in Fiji, which reported prevalence rates of 23.33%, 22.56%, and 18.5%, respectively [43–45]. This could be due to the similar tropical climates, densely populated urban slums and some identical cultural practices between India and Ethiopia. Our pooled estimate was greater than that of a systematic review and meta-analysis conducted globally; a

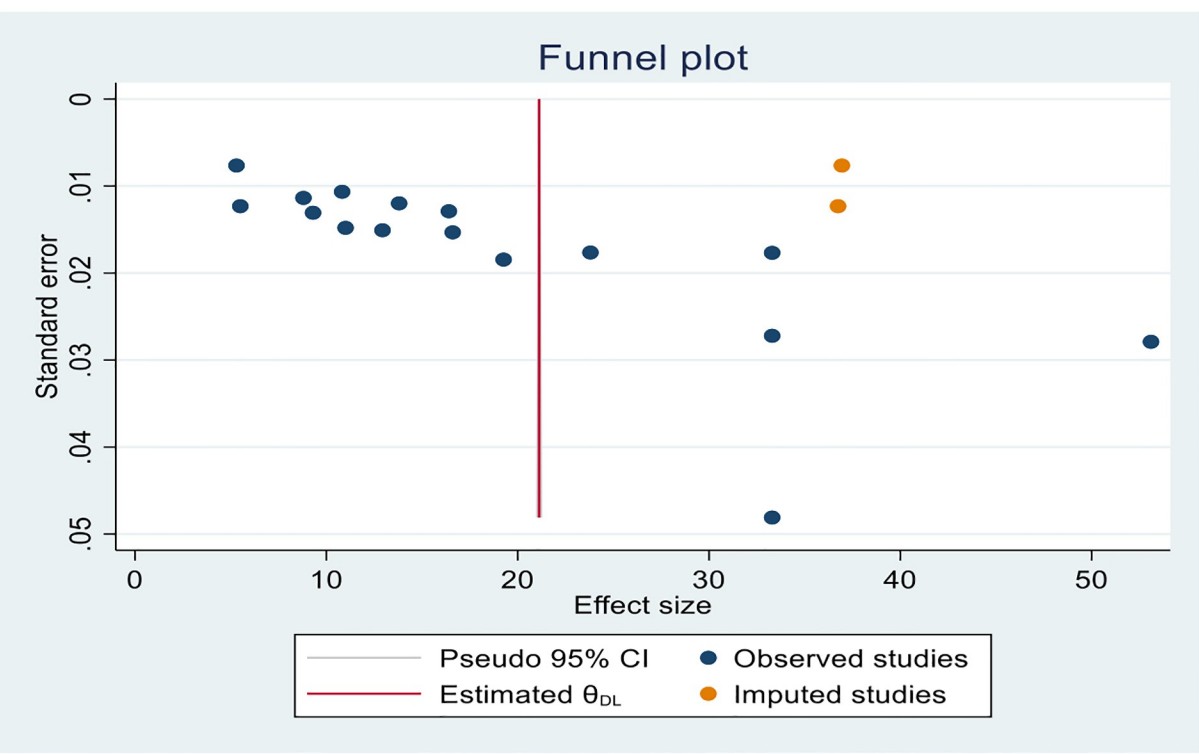

**Fig 10. Funnel plot of eighteen studies after trim-and-fill analysis in the meta-analysis of scabies among school age children in Ethiopia, 2024.**

**Table 2. Factors associated with scabies among school age children in Ethiopia, 2024 were identified.**

| Variable (Reference) | Number of studies | Sample size | Effect size (POR with 95% CI) | Heterogeneity | |
|---|---|---|---|---|---|
| | | | | I² | P-value |
| Sex(female) | 10 | 5992 | 1.44(1.03, 2.00) | 79.5% | 0.000 |
| Father education(literate) | 6 | 3164 | 2.10(1.31, 3.37) | 71.1% | 0.004 |
| Fingernails cut(yes) | 3 | 1767 | 3.26(1.31, 8.10) | 76.8% | 0.013 |
| Family history of scabies(no) | 5 | 2704 | 7.25(4.85, 10.85) | 0.0% | 0.452 |
| Knowledge about scabies(good) | 2 | 1145 | 3.80(2.29, 6.31) | 0.0% | 0.379 |
| Sharing bed(no) | 5 | 3488 | 3.49(2.12, 5.73) | 20.05% | 0.284 |
| Sleep with scabies ill person(no) | 2 | 1085 | 3.49(1.78, 6.83) | 0.0% | 0.892 |
| Frequency of washing cloth(frequent) | 3 | 2297 | 2.39(1.51, 3.77) | 0.0% | 0.420 |
| Sharing cloth(no) | 9 | 4692 | 3.18(1.17, 8.64) | 90.6% | 0.000 |
| Contact with scabies symptom(no) | 6 | 3312 | 5.78(3.73, 8.97) | 43.8% | 0.113 |
| Sleeping place(Bed) | 2 | 1344 | 5.82(2.80, 12.07) | 41.1% | 0.193 |
| Frequency of bath(frequent) | 7 | 3292 | 1.88(1.16, 3.03) | 70.0% | 0.003 |
| Using detergent for washing(yes) | 7 | 3646 | 2.28(1.54, 3.37) | 59.0% | 0.023 |
| Travel to scabies epidemic areas(no) | 3 | 1269 | 2.93(1.33, 6.43) | 63.7% | 0.063 |
| Water source(improved) | 3 | 1535 | 3.04(1.35, 6.86) | 83.6% | 0.002 |

population-based survey conducted in Liberia and Iran reported a prevalence of scabies of 14.0%, 9.3% and 3.1%, respectively [8, 18, 46]. A plausible explanation for the higher pooled prevalence of scabies among school age children in Ethiopia includes certain cultural practices, such as sharing cloth, and warmer, more humid conditions in the country. Our pooled estimate was lower than those reported in cross-sectional surveys conducted in Malaysia, the Solomon Islands, and Timor-Leste, which reported prevalence rates of 31%, 54.3%, and 30.6%, respectively [47, 48]. The reason for this high prevalence could be that Malaysia's warm and humid climate provides ideal conditions for the survival of *Sarcoptes scabiei* mites. Furthermore, limited access to clean water, overcrowded living conditions, the tropical climate in Timor-Leste and favourable environmental conditions for *Sarcoptes scabiei* in the Solomon Islands could contribute to a higher prevalence of scabies. Finally, the large difference in reported prevalence likely comes from the use of various diagnostic methods, such as the IACS, the WHO and national guidelines, leading to inconsistencies in how scabies are identified.

## Associated factors of scabies among school age children in Ethiopia

In addition to examining overall prevalence, our systematic review and meta-analysis also explored factors associated with scabies. Accordingly, sex, education of father, fingernails cut, family history with scabies, knowledge of scabies, sharing a bed, sleeping with scabies ill person, frequency of washing cloth, sharing of cloth, contact with a person who had scabies symptoms, sleeping place, frequency of bath, using a detergent for washing, travel to epidemic areas and water source were the identified predictors for scabies among school age children in Ethiopia.

This meta-analysis of risk factors showed that the odds of scabies among males were 1.44 times higher than females [POR: 1.44; 95% CI (1.03, 2.00)]. This finding is in agreement with studies conducted in American Samoa and Solomon Islands [48, 49] which reported that being male in sex had higher odds of scabies. Males pay less attention to personal hygiene than women. Females are generally more concerned about cleanliness and beauty so they take better care of themselves and maintain cleanliness compared to males [50]. In addition, play behaviour and social interaction patterns of males could contribute to a higher prevalence of scabies. The prevalence of scabies in the present systematic review and meta-analysis was 2.10 times higher among school age children whose fathers were illiterate than their counterparts [POR: 2.10; 95% CI (1.31, 3.37)]. This finding is identical to the study conducted in Egypt and Iran [18, 51]. Studies have found a link between a parent's education level and their ability to implement recommended treatments and preventative measures for scabies in their children [51, 52]. Fathers with higher education levels may be more knowledgeable about scabies and other infectious diseases, including how scabies is transmitted and the importance of early treatment [53].

The likelihood of scabies occurrence was 3.26 times higher among school age children who did not cut their fingernails than those who did cut their fingernails [POR: 3.26; 95% CI (1.31, 8.10)]. This study aligns with research in Indonesia, where scabies rates were lower among schoolchildren who kept their nails trimmed [50]. School age children with a family history of scabies were 7.25 times more likely to encounter scabies than those who did have a family history with scabies [POR: 7.25; 95% CI (4.85, 10.85. This finding is supported by a previous study conducted in Egypt [51]. This could be due to scabies spreading easily through close physical contact and if school age children often have close contact with siblings and parents, so if someone in the family has scabies, the others are at higher risk of catching it [54]. School age children who have poor knowledge about scabies were 3.80 times more likely to have

scabies as compared to their counterparts [POR: 3.80; 95% CI (2.29, 6.31)]. This finding is consistent with a study conducted in Bangladesh [55]. Without awareness of how scabies spread, school age may not take necessary precautions, such as avoiding close contact or washing infected items, facilitating ongoing transmission [56]. The pooled effect of this study indicated that the likelihood of experiencing scabies was increased among school age children who shared a bed [POR:3.49; 95% CI (2.12, 5.73)] and cloth [POR: 3.18; 95% CI (1.17, 8.64)] than their counterpart's. Previous studies from Indonesia [19] have shown a similar association between sharing a bed and cloth with scabies [57]. Scabies mites, the tiny bugs that cause scabies, live on the surface of the skin. This is due to sharing beds or clothes with someone infected with scabies allows the mites to crawl directly from the infected person to the uninfected person [58]. In addition, if one person in a bed-sharing arrangement is infected with scabies, the close quarters can facilitate ongoing transmission and reinfection [59]. The likelihood of scabies occurrence was higher among school age children who slept with a scabies-ill person [POR: 3.49; 95% CI (1.78, 6.83)] and those school age children who had contact with the person who had scabies symptoms [POR: 5.78; 95% CI (3.73, 8.97)] than those who did not. This finding is congruent with the study conducted in Indonesia [60]. Close contact or sleeping with individuals who have scabies is one of the principal ways the condition is transmitted from affected individuals to healthy ones.

The current study reveals that there are significant associations between scabies and school age children who were washing their cloth [POR: 2.39; 95% CI (1.51, 3.77)] and bodies [POR: 1.88; 95% CI (1.16, 3.03)] infrequently. This finding is similar to the study conducted in Pakistan and Cameroon [20, 61]. The longer clothes and body go unwashed, the higher the chance scabies mites have of surviving on the cloth and body. This increases the risk of transmission to anyone who comes into contact. School age children who slept on the floor were 5.82 times more likely to develop scabies as compared to their counterparts [POR: 5.82; 95% CI (2.80, 12.07)]. This finding may relate to the study conducted in Indonesia [19]. This could be due to floors can harbour scabies mites, especially in environments where infected individuals have been this will increase the likelihood of direct contact with these mites. Also, the pooled effect of this study indicated that households who were not using detergents for washing their school age children had a higher likelihood of experiencing scabies than their counterparts [POR: 2.28; 95% CI (1.54, 3.37)]. This finding is congruent with a study conducted in Nigeria and Indonesia [11, 62]. Effective detergents might play a role in eliminating scabies mites, particularly targeting immature stages. Reducing the overall mite burden on the skin can significantly decrease the risk of transmission. School age children who travelled to scabies epidemic areas were 2.93 times more likely to develop scabies than those who did not [POR: 2.93; 95% CI (1.33, 6.43)]. The finding in this study was similar to the study conducted in Poland [63]. Travelling to areas with scabies increases the risk of catching it. This is because close contact, like sharing clothes, playing, or sleeping with someone who has scabies, allows the mites that cause scabies to easily spread from person to person. Finally, the likelihood of scabies was higher among school age children who used unimproved water sources than those who used improved water sources. A systematic review and meta-analysis conducted globally also had similar findings [8]. These water sources may be contaminated with dirt and other substances that can exacerbate skin conditions, making individuals more susceptible to scabies.

### Strengths and limitations of the study

This comprehensive systematic review and meta-analysis provide valuable insights into the pooled prevalence and associated risk factors for scabies among school age children in Ethiopia. However, the study is not without limitations. The exclusive focus on English-language

publications may have resulted in the exclusion of potentially relevant studies conducted in other languages, potentially limiting the strength of the findings. This could lead to an incomplete understanding of the topic, as important data from non-English research could be overlooked. In addition, focusing exclusively on observational studies in a systematic review and meta-analysis may impact the findings due to the inherent limitations of observational data, including various biases (such as selection bias, confounding factors, and recall bias), a lack of experimental rigour compared with randomized controlled trials (RCTs), and the inability to establish causality. As a result, it may overestimate or underestimate the findings. Finally, some of the primary studies had relatively small sample sizes, which may have affected the precision of the estimated pooled prevalence. Importantly, this meta-analysis included data from only five regions of Ethiopia, and the lack of representation from other parts of the country may limit the generalizability of the findings to the entire Ethiopian context. Future research should aim to widen the geographical scope and consider non-English language publications to gain a more comprehensive understanding of the burden of scabies among school-age children across Ethiopia.

## Implications

The high prevalence of scabies among school age children in Ethiopia could lead to significant public health concerns, including increased morbidity, secondary infections, and greater strain on healthcare resources. Economically, the impact of absenteeism due to scabies disrupts children's education and affects long-term development. The social stigma associated with scabies may further isolate affected children, impacting their mental health and social interactions. Additionally, challenges in controlling and preventing scabies emphasize the need for school-focused interventions and educational campaigns to increase awareness, promote hygiene, and address factors contributing to the spread of scabies in this vulnerable age group.

## Conclusions and recommendations

The findings of this comprehensive systematic review and meta-analysis reveal a significantly high prevalence of scabies among school age children in Ethiopia. Several important socio-demographic, environmental, and behavioural factors were identified as being significantly associated with the burden of scabies in this vulnerable population. These include sex, paternal education level, poor nail hygiene, family history of scabies, lack of knowledge about the disease, bed-sharing, contact with individuals with scabies symptoms, infrequent bathing and cloth washing, use of shared water sources, and travel to epidemic areas. Based on our findings, we recommended a multi-pronged approach to address the scabies epidemic among school age children in Ethiopia. Strengthening collaboration between the education, water, and health sectors is crucial for effective scabies prevention and management. This can be achieved by implementing school-based hygiene programs, regular health screenings, and scabies awareness sessions, with support from the local health bureau for treatment and the water sector ensuring clean water and sanitation facilities. Regular cross-sector meetings and training can create a coordinated response, promoting better health for school age children. Furthermore, targeted interventions addressing the identified risk factors, such as promoting proper nail hygiene, cleaning bed materials, and washing clothes with detergent before sharing them with others, should be prioritized. Ethiopia should incorporate scabies management into its universal health coverage package. Given the prevalence of 21%, we recommend mass drug administration with oral ivermectin as an effective and safe strategy to significantly reduce scabies cases. This recommendation aligns with the World Health Organization's guidelines for MDAs in areas with a prevalence of 10% or greater. Additionally, accompanying MDAs with

educational interventions can help increase awareness of scabies transmission, address stigma, and support communities in modifying environmental factors that contribute to its spread. Implementing these evidence-based strategies has the potential to significantly reduce the burden of scabies and improve the health and well-being of school-age children in Ethiopia.

## Supporting information

**S1 File. PRISMA checklist used in the reports of the systematic review and meta-analysis.** (DOCX)

**S2 File. All studies identified in the literature search.** (XLSX)

**S3 File. Newcastle-Ottawa Scale (NOS) quality appraisal/results of the quality assessment of the studies.** (DOCX)

**S4 File. Data set used in generating and analyzing the systematic review and meta-analysis.** (XLSX)

## Author Contributions

**Conceptualization:** Lidetu Demoze, Gelila Yitageasu.

**Data curation:** Lidetu Demoze, Fetlework Gubena, Eyob Akalewold, Gelila Yitageasu.

**Formal analysis:** Lidetu Demoze, Fetlework Gubena, Helen Brhan, Kassaw Chekole Adane, Zemichael Gizaw, Gelila Yitageasu.

**Investigation:** Lidetu Demoze, Tigist Kifle, Gelila Yitageasu.

**Methodology:** Lidetu Demoze, Fetlework Gubena, Kassaw Chekole Adane, Tigist Kifle, Natnael Gizachew, Zemichael Gizaw, Gelila Yitageasu.

**Project administration:** Lidetu Demoze, Kassaw Chekole Adane, Tigist Kifle.

**Resources:** Lidetu Demoze, Natnael Gizachew, Gelila Yitageasu.

**Software:** Lidetu Demoze, Helen Brhan, Kassaw Chekole Adane, Natnael Gizachew, Gelila Yitageasu.

**Supervision:** Lidetu Demoze, Eyob Akalewold, Helen Brhan, Zemichael Gizaw, Amensisa Hailu Tesfaye, Gelila Yitageasu.

**Validation:** Lidetu Demoze, Eyob Akalewold, Helen Brhan, Amensisa Hailu Tesfaye.

**Visualization:** Lidetu Demoze, Eyob Akalewold, Helen Brhan, Amensisa Hailu Tesfaye, Gelila Yitageasu.

**Writing – original draft:** Lidetu Demoze.

**Writing – review & editing:** Lidetu Demoze.

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
