## [Decision Letter · Decision Letter 0]

2 Oct 2024

PONE-D-24-23061Prevalence and associated factors of scabies among school age children in Ethiopia: A systematic review and meta-analysisPLOS ONE

Dear Dr. Demoze,

Thank you for submitting your manuscript to PLOS ONE. After careful consideration, we feel that it has merit but does not fully meet PLOS ONE’s publication criteria as it currently stands. Therefore, we invite you to submit a revised version of the manuscript that addresses the points raised during the review process. In addition to addressing the technical comments from the reviewers, English language expression needs a thorough checking. The experts (especially reviewer 2) have given a comprehensive list of suggestions for improvement, and authors may use them accordingly.

We look forward to receiving your revised manuscript.

Kind regards,

Muhammad Farooq Umer, PhD Epidemiology and Health Statistics

Academic Editor

PLOS ONE

Journal Requirements:

-https://doi.org/10.1371/journal.pone.0211764

-https://doi.org/10.1186/s12891-023-06633-1

In your revision ensure you cite all your sources (including your own works), and quote or rephrase any duplicated text outside the methods section. Further consideration is dependent on these concerns being addressed.

3. As required by our policy on Data Availability, please ensure your manuscript or supplementary information includes the following: A numbered table of all studies identified in the literature search, including those that were excluded from the analyses. For every excluded study, the table should list the reason(s) for exclusion. If any of the included studies are unpublished, include a link (URL) to the primary source or detailed information about how the content can be accessed. A table of all data extracted from the primary research sources for the systematic review and/or meta-analysis. The table must include the following information for each study: Name of data extractors and date of data extraction Confirmation that the study was eligible to be included in the review. All data extracted from each study for the reported systematic review and/or meta-analysis that would be needed to replicate your analyses. If data or supporting information were obtained from another source (e.g. correspondence with the author of the original research article), please provide the source of data and dates on which the data/information were obtained by your research group. If applicable for your analysis, a table showing the completed risk of bias and quality/certainty assessments for each study or outcome. Please ensure this is provided for each domain or parameter assessed. For example, if you used the Cochrane risk-of-bias tool for randomized trials, provide answers to each of the signalling questions for each study. If you used GRADE to assess certainty of evidence, provide judgements about each of the quality of evidence factor. This should be provided for each outcome. An explanation of how missing data were handled. This information can be included in the main text, supplementary information, or relevant data repository. Please note that providing these underlying data is a requirement for publication in this journal, and if these data are not provided your manuscript might be rejected.

Reviewers' comments:

Reviewer's Responses to Questions

**Comments to the Author**

1. Is the manuscript technically sound, and do the data support the conclusions?

Reviewer #1: Yes

Reviewer #2: Partly

2. Has the statistical analysis been performed appropriately and rigorously? 

Reviewer #1: Yes

Reviewer #2: Yes

3. Have the authors made all data underlying the findings in their manuscript fully available?

Reviewer #1: Yes

Reviewer #2: Yes

4. Is the manuscript presented in an intelligible fashion and written in standard English?

Reviewer #1: Yes

Reviewer #2: No

5. Review Comments to the Author

Reviewer #1: The present is an interesting systematic review

Some issues should be added

1) A lot of predictors have been emdedded. I think the most important ones should be appraised, from the importance of predictors. (see PMID: 22360945)

2) it is not clear if authors pooled predictors together or not

Reviewer #2: The title of the paper is clear but could be enhanced by adding a phrase to emphasize its public health significance. For example, "Burden and Determinants of Scabies in Ethiopian School-Age Children: A Systematic Review and Meta-Analysis with Public Health Implications" would make the study’s impact more apparent.

The abstract is generally well-structured, providing key information on the background, methods, results, and conclusions. However, the background section could benefit from additional context on why the review was necessary. You could mention the lack of comprehensive data on scabies prevalence in Ethiopia and its global health implications. In the methods section, clarify the study period covered and specify the databases searched. In the results section, highlight the strongest associated factors, and the conclusion could include a more specific public health recommendation, such as the need for coordinated interventions between the healthcare and education sectors.

In the introduction, you provide a good explanation of scabies as a public health issue. It might help to include more details on the global burden of scabies and comparisons with other neglected tropical diseases. When mentioning previous studies, briefly summarize their prevalence ranges to emphasize the variability in the findings and the need for this meta-analysis. The introduction could also more explicitly highlight the novelty of this review compared to existing studies.

The methods section is thorough, following PRISMA guidelines, which is commendable. In the search strategy, consider providing more details on the rationale for choosing the specific databases and how the Google Scholar search was manually conducted. The inclusion and exclusion criteria are mostly clear, but you could elaborate on the reasons for excluding conference abstracts and non-peer-reviewed sources. It would also help to expand on the statistical rationale for using a random-effects model, especially in light of the study heterogeneity, and how the sensitivity analysis was conducted.

The results section is well organized, and the quantitative analysis is strong. However, the data presentation could be more engaging by including a brief narrative summary before diving into the statistical details. For example, introduce the study characteristics and then transition into the statistical results. The figures, while informative, may require enhancement. Ensure that all figures are presented in high resolution (300 DPI or higher) and that the color schemes are accessible for both color-blind and grayscale viewers. Descriptive labels and legends should be added to figures to improve clarity and make them easier to interpret without needing to refer back to the text. Additionally, it would be useful to include a detailed explanation for significant findings in a more digestible format, such as simplified explanations of the key associations and their implications.

In the discussion, you effectively relate your findings to previous research, but this section could be further strengthened by expanding on comparisons with studies from other countries or regions with similar climates. Additionally, more explanation is needed on the mechanisms behind the associated factors, such as why father’s education level or bed-sharing is strongly linked to scabies. Adding context on biological, social, or cultural mechanisms will deepen the analysis. The limitations section is good, but you could elaborate on how the language and publication biases, or the focus on observational studies, may have impacted the findings. Discussing potential over- or under-estimation of prevalence due to study methods would also be insightful. It is crucial to emphasize how these limitations might affect the generalizability of your findings. Furthermore, the public health implications of your findings should be expanded. Discuss specific interventions or policy recommendations, such as implementing school-based hygiene programs or increasing collaboration between healthcare and educational institutions.

The conclusion summarizes the study well but could be more impactful. Provide actionable recommendations for stakeholders, such as targeted education campaigns or infrastructural improvements to sanitation in schools. Strengthening the collaboration between the education, water, and health sectors is an important recommendation, and you should emphasize how this could be implemented in practice.

Regarding the figures and tables, ensure that they meet publication quality standards. The resolution of all figures should be high, ideally 300 DPI or more. Visual aids, such as forest plots or funnel plots, need to be well-labeled and easily interpretable. Additionally, maintain a consistent visual style across all figures, including uniform font sizes, color schemes, and formatting.

Finally, from a technical perspective, ensure that the manuscript adheres to the journal’s citation style requirements. There are a few sentences throughout that could be revised for clarity and flow by simplifying their structure. For example, the sentence "Infectious skin diseases and infestations like scabies are a common problem in school age children owing to close contact between classmates" could be revised to "Scabies and other infectious skin diseases are common among school-age children due to close contact between classmates."

In terms of structure, the overall organization of the manuscript is logical, but the transitions between sections could be smoother. For example, transitioning between results and discussion with a brief summary of key findings before diving into the discussion would improve readability. Additionally, targeting a journal with a focus on public health or tropical medicine will ensure that the paper reaches the appropriate audience. Journals such as PLOS Neglected Tropical Diseases or the International Journal of Dermatology may be good fits for this work.

To prepare for submission, ensure that the figures are of high quality, the manuscript is proofread for grammatical errors, and that the formatting adheres to the journal’s guidelines.

6. PLOS authors have the option to publish the peer review history of their article (what does this mean?). If published, this will include your full peer review and any attached files.

Reviewer #1: **Yes: **Fabrizio D'Ascenzo

Reviewer #2: No

---

## [Author Response · Author response to Decision Letter 0]

1 Nov 2024

Responses to the Editors and review’s comments

Dear PLOS ONE editorial team,

Thank you for giving us the opportunity to submit a revised draft of the manuscript, and we would also like to thank you for your crucial comments on our paper (Manuscript ID: PONE-D-24-23061). We are very concerned and have combined all the suggested comments provided, which we believe strengthen our paper, and we hope this will render our paper eligible for consideration for publication in your reputed journal. We appreciate the time and effort that you and the reviewers dedicated to providing feedback on our manuscript and are grateful for the insightful comments and valuable improvements to our paper for publication.

The authors would like to inform you that we have addressed the comments and recommendations of the handling editor point by point. In addition, throughout our revision, we made our best corrections too. All changes made to the original version are highlighted using tracking changes and attached as “Revised Manuscript with Track Changes”. The unmarked copy of the manuscript is also attached as “Manuscript”. In addition, please see below a rebuttal letter that responds to each point raised by the handling editor, and this letter is also attached to the submission as “Response to Reviewers”.

Response to editor’s comments

Comments from the handling editor:

Author’s response: Dear Editor, thank you very much for your recommendation. We have made the corrections accordingly to meet the journal requirements. 

-https://doi.org/10.1371/journal.pone.0211764

-https://doi.org/10.1186/s12891-023-06633-1

In your revision ensure you cite all your sources (including your own works), and quote or rephrase any duplicated text outside the methods section. Further consideration is dependent on these concerns being addressed.

Author’s response: Dear Editor, thank you very much for your recommendations. We have made the necessary corrections to meet the journal's requirements. However, there are certain terms, such as "systematic," "review," "meta-analysis," and "prevalence," that are unavoidable when conducting a systematic review and meta-analysis of prevalence studies, which may account for some similarity with the mentioned papers.

3. As required by our policy on Data Availability, please ensure your manuscript or supplementary information includes the following: A numbered table of all studies identified in the literature search, including those that were excluded from the analyses. For every excluded study, the table should list the reason(s) for exclusion. If any of the included studies are unpublished, include a link (URL) to the primary source or detailed information about how the content can be accessed. A table of all data extracted from the primary research sources for the systematic review and/or meta-analysis. The table must include the following information for each study: Name of data extractors and date of data extraction Confirmation that the study was eligible to be included in the review. All data extracted from each study for the reported systematic review and/or meta-analysis that would be needed to replicate your analyses. If data or supporting information were obtained from another source (e.g. correspondence with the author of the original research article), please provide the source of data and dates on which the data/information were obtained by your research group. If applicable for your analysis, a table showing the completed risk of bias and quality/certainty assessments for each study or outcome. Please ensure this is provided for each domain or parameter assessed. For example, if you used the Cochrane risk-of-bias tool for randomized trials, provide answers to each of the signalling questions for each study. If you used GRADE to assess certainty of evidence, provide judgements about each of the quality of evidence factor. This should be provided for each outcome. An explanation of how missing data were handled. This information can be included in the main text, supplementary information, or relevant data repository. Please note that providing these underlying data is a requirement for publication in this journal, and if these data are not provided your manuscript might be rejected.

Author’s response: Dear Editor, thank you very much for your recommendation. We have made the corrections accordingly to meet the journal requirements. 

Comments from Reviewer #1: 

A lot of predictors have been emdedded. I think the most important ones should be appraised, from the importance of predictors. (see PMID: 22360945) 2) it is not clear if authors pooled predictors together or not. 

Author’s response: Dear Reviewer, thank you very much for your suggestions. However, all the included predictors are essential in determining the factors influencing scabies prevalence among school-age children, and each predictor is distinct. While we have pooled the predictors together, we can provide the Stata output as a supplementary file if you would like to review it.

Comments from Reviewer #2: 

The title of the paper is clear but could be enhanced by adding a phrase to emphasize its public health significance. For example, "Burden and Determinants of Scabies in Ethiopian School-Age Children: A Systematic Review and Meta-Analysis with Public Health Implications" would make the study’s impact more apparent.

Author’s response: Dear Reviewer, thank you very much for your suggestions. We have made the corrections accordingly. 

The abstract is generally well-structured, providing key information on the background, methods, results, and conclusions. However, the background section could benefit from additional context on why the review was necessary. You could mention the lack of comprehensive data on scabies prevalence in Ethiopia and its global health implications. In the methods section, clarify the study period covered and specify the databases searched. In the results section, highlight the strongest associated factors, and the conclusion could include a more specific public health recommendation, such as the need for coordinated interventions between the healthcare and education sectors.

Author’s response: Dear Reviewer, thank you very much for your recommendation. We have made the corrections accordingly. 

In the introduction, you provide a good explanation of scabies as a public health issue. It might help to include more details on the global burden of scabies and comparisons with other neglected tropical diseases. When mentioning previous studies, briefly summarize their prevalence ranges to emphasize the variability in the findings and the need for this meta-analysis. The introduction could also more explicitly highlight the novelty of this review compared to existing studies.

Author’s response: Dear Reviewer, thank you very much for your recommendation. We have made the corrections accordingly. 

The methods section is thorough, following PRISMA guidelines, which is commendable. In the search strategy, consider providing more details on the rationale for choosing the specific databases and how the Google Scholar search was manually conducted. The inclusion and exclusion criteria are mostly clear, but you could elaborate on the reasons for excluding conference abstracts and non-peer-reviewed sources. It would also help to expand on the statistical rationale for using a random-effects model, especially in light of the study heterogeneity, and how the sensitivity analysis was conducted.

Author’s response: Dear Reviewer, thank you very much for your recommendation. We have made the corrections accordingly. 

The results section is well organized, and the quantitative analysis is strong. However, the data presentation could be more engaging by including a brief narrative summary before diving into the statistical details. For example, introduce the study characteristics and then transition into the statistical results. The figures, while informative, may require enhancement. Ensure that all figures are presented in high resolution (300 DPI or higher) and that the color schemes are accessible for both color-blind and grayscale viewers. Descriptive labels and legends should be added to figures to improve clarity and make them easier to interpret without needing to refer back to the text. Additionally, it would be useful to include a detailed explanation for significant findings in a more digestible format, such as simplified explanations of the key associations and their implications.

Author’s response: Dear Reviewer, thank you very much for your suggestions. We begin by introducing the study characteristics, followed by a transition into the statistical results, including subgroup analysis, sensitivity analysis, publication bias assessment, and other relevant analyses. All figures are presented in high resolution (300 DPI or higher), with color schemes accessible to both color-blind and grayscale viewers. Descriptive labels and legends are also included. Additionally, significant findings are explained in a detailed and accessible format.

In the discussion, you effectively relate your findings to previous research, but this section could be further strengthened by expanding on comparisons with studies from other countries or regions with similar climates. Additionally, more explanation is needed on the mechanisms behind the associated factors, such as why father’s education level or bed-sharing is strongly linked to scabies. Adding context on biological, social, or cultural mechanisms will deepen the analysis. The limitations section is good, but you could elaborate on how the language and publication biases, or the focus on observational studies, may have impacted the findings. Discussing potential over- or under-estimation of prevalence due to study methods would also be insightful. It is crucial to emphasize how these limitations might affect the generalizability of your findings. Furthermore, the public health implications of your findings should be expanded. Discuss specific interventions or policy recommendations, such as implementing school-based hygiene programs or increasing collaboration between healthcare and educational institutions.

Author’s response: Dear Reviewer, thank you very much for your recommendation. We have made the corrections accordingly. 

The conclusion summarizes the study well but could be more impactful. Provide actionable recommendations for stakeholders, such as targeted education campaigns or infrastructural improvements to sanitation in schools. Strengthening the collaboration between the education, water, and health sectors is an important recommendation, and you should emphasize how this could be implemented in practice.

Author’s response: Dear Reviewer, thank you very much for your recommendation. We have made the corrections accordingly. 

Regarding the figures and tables, ensure that they meet publication quality standards. The resolution of all figures should be high, ideally 300 DPI or more. Visual aids, such as forest plots or funnel plots, need to be well-labeled and easily interpretable. Additionally, maintain a consistent visual style across all figures, including uniform font sizes, color schemes, and formatting.

Author’s response: Dear Reviewer, thank you very much for your recommendation. We have reviewed and addressed all of the issues mentioned above.

Finally, from a technical perspective, ensure that the manuscript adheres to the journal’s citation style requirements. There are a few sentences throughout that could be revised for clarity and flow by simplifying their structure. For example, the sentence "Infectious skin diseases and infestations like scabies are a common problem in school age children owing to close contact between classmates" could be revised to "Scabies and other infectious skin diseases are common among school-age children due to close contact between classmates."

Author’s response: Dear Reviewer, thank you very much for your recommendation. We have reviewed and addressed all of the issues mentioned above.

In terms of structure, the overall organization of the manuscript is logical, but the transitions between sections could be smoother. For example, transitioning between results and discussion with a brief summary of key findings before diving into the discussion would improve readability. Additionally, targeting a journal with a focus on public health or tropical medicine will ensure that the paper reaches the appropriate audience. Journals such as PLOS Neglected Tropical Diseases or the International Journal of Dermatology may be good fits for this work.

Author’s response: Dear Reviewer, thank you very much for your recommendation. We have made the corrections accordingly. We will consider the recommended journals for future works.

To prepare for submission, ensure that the figures are of high quality, the manuscript is proofread for grammatical errors, and that the formatting adheres to the journal’s guidelines.

Author’s response: Dear Reviewer, thank you very much for your recommendation. We have made the corrections accordingly.

---

## [Decision Letter · Decision Letter 1]

19 Nov 2024

Burden and Determinants of Scabies in Ethiopian School Age Children: A Systematic Review and Meta-Analysis with Public Health Implications

PONE-D-24-23061R1

Dear Dr. Demoze,

We’re pleased to inform you that your manuscript has been judged scientifically suitable for publication and will be formally accepted for publication once it meets all outstanding technical requirements.

**It is advised to improve the DPIs of images and figures so that the publication process is smoothly handled.**

Kind regards,

Muhammad Farooq Umer, PhD Epidemiology and Health Statistics

Academic Editor

PLOS ONE

Additional Editor Comments (optional):

Reviewers' comments:

Reviewer's Responses to Questions

**Comments to the Author**

1. If the authors have adequately addressed your comments raised in a previous round of review and you feel that this manuscript is now acceptable for publication, you may indicate that here to bypass the “Comments to the Author” section, enter your conflict of interest statement in the “Confidential to Editor” section, and submit your "Accept" recommendation.

Reviewer #1: All comments have been addressed

Reviewer #2: All comments have been addressed

2. Is the manuscript technically sound, and do the data support the conclusions?

Reviewer #1: Yes

Reviewer #2: Yes

3. Has the statistical analysis been performed appropriately and rigorously? 

Reviewer #1: Yes

Reviewer #2: Yes

4. Have the authors made all data underlying the findings in their manuscript fully available?

Reviewer #1: Yes

Reviewer #2: Yes

5. Is the manuscript presented in an intelligible fashion and written in standard English?

Reviewer #1: Yes

Reviewer #2: Yes

6. Review Comments to the Author

Reviewer #1: All comments have been addressed and authors should be congratulated for such a relevant paper to be published.

Reviewer #2: nothing more to add, just improve the quality of the images and figures.

good work, the current form of the manuscript is feasible for publication.

7. PLOS authors have the option to publish the peer review history of their article (what does this mean?). If published, this will include your full peer review and any attached files.

Reviewer #1: **Yes: **Fabrizio D'Ascenzo

Reviewer #2: **Yes: **Fausto Cabezas-Mera

---

## [Editor Report · Acceptance letter]

22 Nov 2024

PONE-D-24-23061R1 

PLOS ONE

Dear Dr. Demoze, 

I'm pleased to inform you that your manuscript has been deemed suitable for publication in PLOS ONE. Congratulations! Your manuscript is now being handed over to our production team.

Kind regards, 

on behalf of

Dr. Muhammad Farooq Umer 

Academic Editor

PLOS ONE